# Extreme Interocular Asymmetry in an Atypical Case of a Hydroxychloroquine-Related Retinopathy

**DOI:** 10.3390/medicina58070967

**Published:** 2022-07-21

**Authors:** Gabriel Hallali, Zari Seyed, Anne-Véronique Maillard, Karima Drine, Laurence Lamour, Céline Faure, Isabelle Audo

**Affiliations:** 1Ophthalmic Pediatric Unit, Centre Hospitalier National d’Ophtalmologie des Quinze-Vingts, 28 rue de Charenton, F-75012 Paris, France; 2Electrophysiology Unit, Centre Hospitalier National d’Ophtalmologie des Quinze-Vingts, 28 rue de Charenton, F-75012 Paris, France; zseyed@15-20.fr (Z.S.); avmaillard@15-20.fr (A.-V.M.); kdrime@15-20.fr (K.D.); lamour@15-20.fr (L.L.); celinefaureoph@gmail.com (C.F.); 3Centre Hospitalier National d’Ophtalmologie des Quinze-Vingts, Centre de Référence Maladies Rares REFERET et DHU Sight Restore, INSERM-DGOS CIC1423, 28 rue de Charenton, F-75012 Paris, France; 4Hôpital Privé Saint Martin, Ramsay Générale de Santé, 18 rue des Roquemonts, 14000 Caen, France; 5Sorbonne Université, Institut National de la Santé Et de la Recherche Médicale (INSERM), Centre National de la Recherche Scientifique (CNRS), Institut de la Vision, 17 rue Moreau, F-75012 Paris, France

**Keywords:** hydroxychloroquine, hydroxychloroquine toxicity screening, bull’s eye maculopathy, parafoveal retinal hydroxychloroquine toxicity, pericentral retinal hydroxychloroquine toxicity

## Abstract

*Background and Objectives*: Long-term hydroxychloroquine (HCQ) therapy can lead to retinal toxicity. Typically, it is characterized by a bull’s eye maculopathy. More recently, a “pericentral” form of HCQ retinopathy that predominantly affects patients of Asian descent has been described. To our knowledge, this is the first reported case where such an asymmetry between the right and the left eye in the toxicity profile is observed. *Case presentation*: The patient presented with a 12-year exposure to HCQ at a daily dose of 4.35 mg/kg. She presented an inferior pericentral-only phenotype of HCQ toxicity on the right eye and a perifoveal-only toxicity on the left eye. Modest progression of toxicity was observed on both eyes over the seven years of follow-up, despite drug discontinuation. *Conclusions*: To our knowledge, this is the first time that two different phenotypes of HCQ-related retinopathy are found in the same patient, challenging our understanding of the pathophysiology of HCQ retinal toxicity.

## 1. Introduction

Long-term hydroxychloroquine (HCQ) therapy can lead to retinal toxicity [1]. This retinal alteration has been typically characterized as a photoreceptor layer thinning that begins as a parafoveal ring and progresses over time to become a “bull’s eye” maculopathy, an appearance which is caused by the perifoveal atrophy with foveal sparing. More recently, a “pericentral” form of HCQ retinopathy that predominantly affects patients of Asian descent has been described [2]. Pericentral is defined as an outer retinal loss more than 7° from the fovea. Although it is reported more commonly in Asian patients, this can also be seen in other ethnic backgrounds [3]. A mixed pattern where there is evidence of pericentral and perifoveal toxicity has also been described in the same eye [4].

## 2. Case Presentation

A 50-year-old Chinese female was referred to our electrophysiology unit for HCQ toxicity screening in 2013. She complained of progressive visual field loss in her left eye. She had a history of systemic lupus erythematosus that was managed by a daily dose of 200 mg (4.35 mg/kg) of HCQ for approximately 14 years. Ophthalmic examination at the time of the initial drug prescription was reported as normal and she admitted no subsequent ophthalmic screening. Her height and weight were 156 cm and 46 kg, respectively, with a body mass index of 16.4. She had no other relevant medical history and, more specifically, she had normal renal and hepatic functions. A daily dose of 5 mg of prednisolone was also taken by the patient.

On presentation, the patient’s best-corrected visual acuity was 20/20 and 20/25 in the right and left eye, respectively. Her optimal refraction was +1.75(−0.5)130° for the right eye and +0.5(−0.5)50° for the left eye. A multifocal electroretinogram (mfERG) was recorded binocularly after pupil dilation using Dawson, Trick, and Litzkow (DTL) corneal electrodes and an LCD monitor (Diagnosys LLC, Lowell, MA, USA), according to the International Society for Clinical Electrophysiology of Vision (ISCEV) recommendations [5]. This revealed decreased peripheral responses inferiorly in the right eye, while the pericentral responses were altered in the second and third ring of the left eye (Figure 1A). The 10-2 Sita fast Humphrey perimetry showed a peripheral supero-nasal scotoma in the right eye, while a paracentral horse-shoe-shaped scotoma around the 2.5° superior field was noted in the left eye (Figure 1B).

A full-field ERG (ff-ERG) was recorded according to the ISCEV [6] on an Espion E^3^ system (Dyagnosis LLC, Cambridge, United Kingdom). Scotopic responses were within the normal limits, whereas photopic responses had decreased amplitudes in the right eye with no implicit time shift, in keeping with a unilateral restricted cone-system dysfunction (Figure 2). The anterior segments were unremarkable. A fundus examination revealed normal optic discs and no vitreous cell or arterial attenuation (Figure 3A,E). Peripheral retinal atrophy was seen inferiorly in the right eye, while the left macula showed subtle perifoveal atrophy.

The 55° fundus autofluorescence imaging (Heidelberg Retina Angiograph 2 (HRA2), Heidelberg Engineering, Heidelberg, Germany) demonstrated a large area of decreased autofluorescence following the inferior temporal vascular arcade, which was delimited by a hyper-autofluorescence line in the right eye (Figure 3C). No macular autofluorescence abnormality was noted (Figure 3B). The autofluorescence imaging of the left eye showed a half ring of decreased autofluorescence that was surrounded by a ring of increased autofluorescence shaped as a “bull’s eye” maculopathy (Figure 3F,G). Spectral domain optical coherence tomography (SD-OCT) (Heidelberg Spectralis, Heidelberg Engineering) imaging (Figure 3D) demonstrated a loss of the outer retinal hyperreflective bands inferotemporally in the right eye. The left eye showed a parafoveal loss of outer retinal hyperreflective bands with foveal sparing correlated with autofluorescence changes (Figure 3H). Considering the appearance of the right eye, a diagnosis of hydroxychloroquine toxicity was presumed and HCQ usage was immediately stopped. A seven-year follow-up showed a modest and slow progression of these structural alterations, despite drug discontinuation (Figure 4A), (Figure 4B,C), as already reported in other cases [7].

## 3. Discussion

Parafoveal and pericentral retinal HCQ toxicity has been well reported [2,4,8], and our patient, with a dosage of 4.35 mg/kg per day for 14 years and a low weight, had an increased risk of developing macular alterations of about 0.2 [1]. We presume that the retina abnormalities that are observed in this patient are due to HCQ toxicity. To our knowledge, this is the first time that such an asymmetry of the toxicity phenotype is described in the same patient. We were not aware of any predisposing factors that led to this unusual presentation in our patient. A differential diagnosis such as an infectious or inflammatory related process could be discussed, although the patient did not report any previous peculiar ophthalmic history besides her recent symptoms. Furthermore, there is no pigmented demarcation line or linear subretinal fibrosis in the RE, as encountered in chronic retinal detachment. The probability that an unnoticed former exudative choroidal detachment created the right eye abnormalities is low but still a possibility.

Inflammatory diseases (e.g., systemic lupus erythematosus (SLE) or antiphospholipid syndrome) and thromboembolic diseases (e.g., disseminated intravascular coagulation (DIC) and thrombocytopenic purpura) can impair choroidal vascular flow with subsequent serous retinal detachment [9,10]. However, no Elschnig spot or triangular RPE alteration match with the pattern of the choroidal vasculature [11]. Vogt-Koyanagi-Harada disease and posterior scleritis could also have been possible causes of previous serous retinal detachment. However, exudative detachment is strongly linked with inflammation severity and our patient did not report any history of ocular pain, or any auditory, skin and neurologic signs [12].

In the past, our patient took a high dose of corticosteroid for severe inflammation that was secondary to SLE, and a past bullous serous retinal detachment cannot be excluded, as corticosteroids are a well-known risk factor for central serous chorioretinopathy [13].

Moreover, type III uveal effusion syndrome (defined as having neither nanophtalmia nor scleral thickening) has recently been found to be associated with pachychoroidopathy spectrum disease [14,15], but this condition is very rare and the semiology does not fit this case.

Also intriguing is the absence of implicit time shift in the FF-ERG, suggesting a localized retinal dysfunction which does not expand to the entire retina in the right eye. FF-ERG abnormalities are not well documented in the literature in pericentral retinal HCQ toxicity. This absence of implicit time shift may be specific to such toxicity, unlike the delay that is reported in parafoveal toxicity [16], or link to another differential diagnosis, as mentioned above.

The pathophysiology of HCQ toxicity remains unknown. The 2016 American Academy Ophtalmology (AAO) recommendations were defined by relying on a large cohort study of 2361 patients’ major risk factors for HCQ maculopathy [1]. These risk factors include an HCQ dose of >5.0 mg/kg, a treatment duration that is superior to 5 years, renal dysfunction, tamoxifen intake, and preexisting macular disease [7]. Cases with major risk factors for retinal toxicity should be clearly identified and monitored appropriately.

It is also now well recognized that Asian patients present more commonly with a pericentral toxicity that may suggest the effect of the genetic factors [2]. To date, there are three main phenotypes of HCQ toxicity reported in the literature: parafoveal HCQ toxicity, which resembles a pseudo-bull’s eye maculopathy that is predominant inferiorly; pericentral retinopathy that is mainly reported in patients of Asian origin but can also happen in patients of Caucasian origin; and a severe phenotype that associates a parafoveal and pericentral phenotypes.

We suggest that patients of Asian origin with significant risks of HCQ toxicity undergo a bi-annual screening of HCQ toxicity, including a Humphrey 10-2 and large SD-OCT vertical scans. Examiners must look critically at the edge of 10-2 fields. When possible, widefield FAF should be obtained, and 30-2 fields may be indicated in addition to 10-2 fields, as recommended by Melles & al [2]. Polymorphic variants on the *ABCA4* gene have been suggested to increase or decrease the risk of HCQ-related maculopathy [17,18], even though the impact of such variants remains elusive. The strong asymmetry of our case pleads against the genetic hypothesis. Indeed, it is hard to understand why only the pericentral cells of the right eye were sensible to HCQ, while only the perifoveal cells were damaged on the left eye. It is noteworthy that an increased risk of HCQ retinopathy related to fundus pigmentation also seems unlikely, because highly pigmented patients of African descent with pericentral retinopathy is uncommon. In our case, the fundus did not show any asymmetry in normal choroidal or retinal pigment epithelium pigmentation. Experimental studies show that HCQ binds to melanin in RPE cells, but it is unclear if this binding triggers drug toxicity, or provides a protective mechanism to remove the drug from the outer retina [7]. A recent in vitro study on RPE cells suggests that these cells contribute to the blood–retinal barrier to HCQ [19]. Local defects of the blood–retinal barrier or abnormality of the choroidal vascularization could be explored. It is noteworthy that hemodynamic differences were reported in patients with asymmetric age-related macular degeneration [20]. A recent article reported an asymmetric case of HCQ toxicity in a 62-year-old White woman with erosive rheumatoid arthritis and tried to address the challenges of understanding HCQ asymmetric retinal toxicity. The woman developed a hydroxychloroquine toxicity in her phakic eye, with her aphakic fellow eye only mildly affected [21]. Authors attributed this asymmetry to optical factors. In our case, the optical factor cannot explain the asymmetry. As discussed above, the classic genetic hypothesis or pigmentation asymmetry can also be ruled out to explain our case. We wonder if intraocular blood flow dynamics should be investigated.

## 4. Conclusions

Our case emphasizes the importance of a specific screening that includes an SD-OCT wide scan and 24-2 visual field to detect all the phenotypes of HCQ retinopathy. The importance of a close follow-up of patients under hydroxychloroquine amid the COVID-19 pandemic must be stressed.

## Figures and Tables

**Figure 1 medicina-58-00967-f001:**
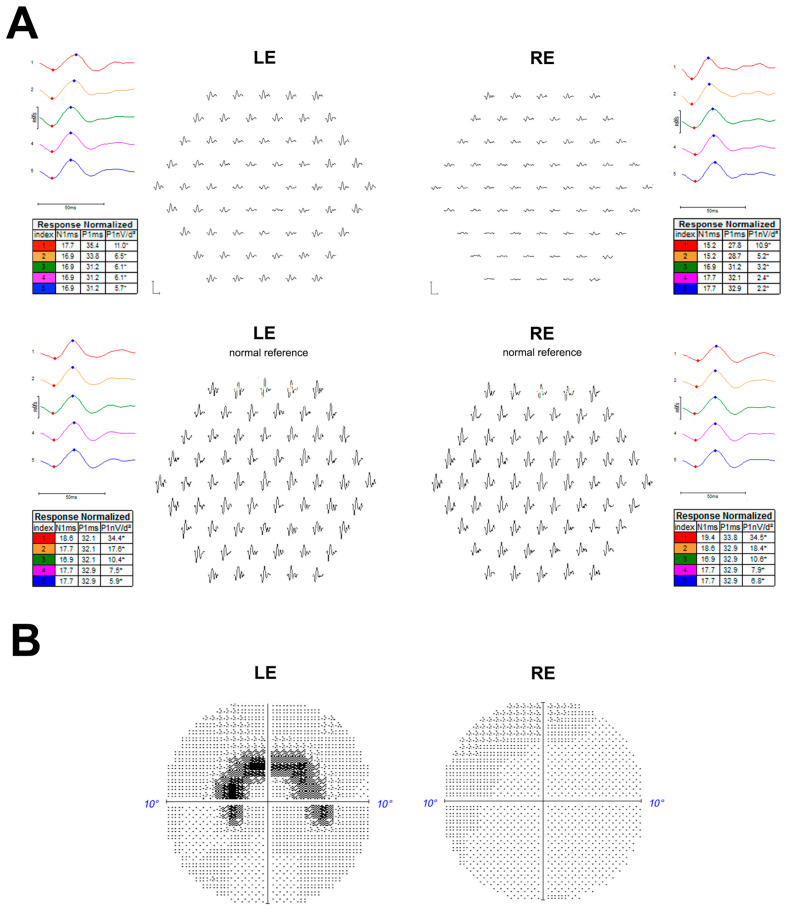
Multifocal Electroretinogram (ERG) (graphs and ring ratios) recorded with Dawson, Trick, and Litzkow (DTL) electrodes on a Diagnosys^®^ system, (Dyagnosis LLC, Cambridge, United Kingdom (2020) (traces from the patient are presented in the upper part with responses from a normal subject in the lower part on retinal view display) with decreased pericentral responses in the second and third ring of the left eye and decreased responses in the fourth and fifth ring in the right eye (**A**). Left and right eye 10-2 Swedish interactive thresholding algorithm (SITA) fast Humphrey visual field, respectively, showing paracentral horse-shoe-shaped scotoma and peripheral supero-nasal campimetric alteration (**B**). LE = Left eye; RE = Right eye.

**Figure 2 medicina-58-00967-f002:**
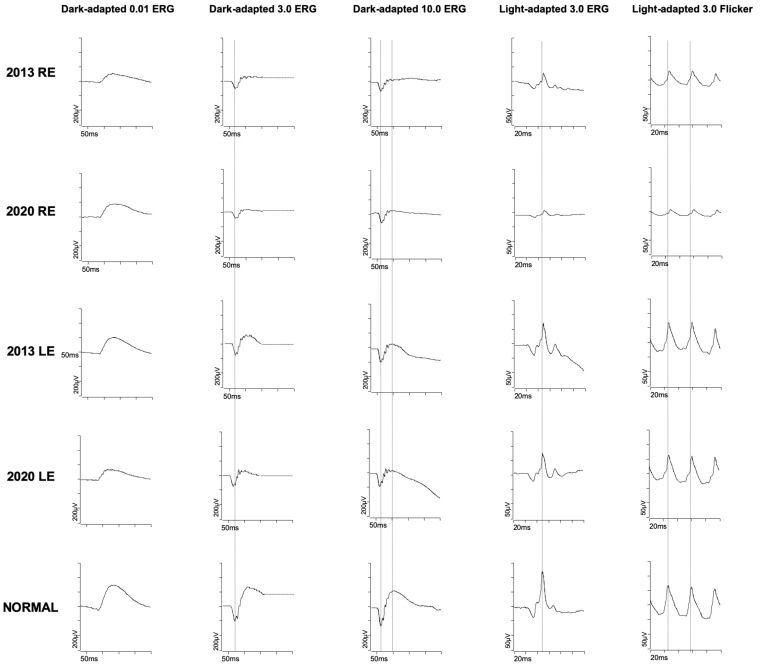
Full-field Electroretinogram (ERG), left and right eye at presentation and after seven years displaying decreased amplitudes of the photopic responses of the right eye that worsened with follow-up. Scotopic responses were within normal limits.

**Figure 3 medicina-58-00967-f003:**
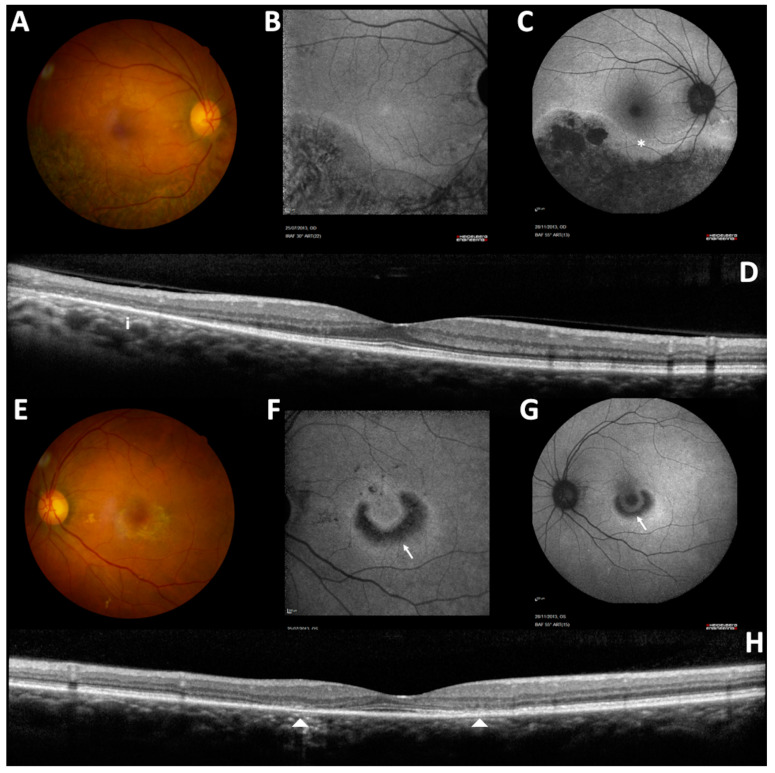
Left and right eye imaging at presentation: color fundus (**A**,**E**); 32° (**B**,**F**); and 55° (**C**,**G**) fundus autofluorescence imaging and spectral-domain optical coherence tomography (OCT) (macular vertical scan) (**D**,**H**). The RE showed pericentral hydroxychloroquine (HCQ) toxicity pattern with an atrophic hypoautofluorescent area along the inferior temporal vascular arcade delimited by a hyperautofluorescent line (*). The LE showed a perifoveal HCQ toxicity pattern with a typical “bull’s eye maculopathy” (arrow). Vertical SD-OCT scans demonstrated loss of the external limiting membrane, ellipsoid zone and interdigitation zone along the inferior temporal vascular arcade in the RE (**i**) and in the parafoveal area in the LE (arrow heads).

**Figure 4 medicina-58-00967-f004:**
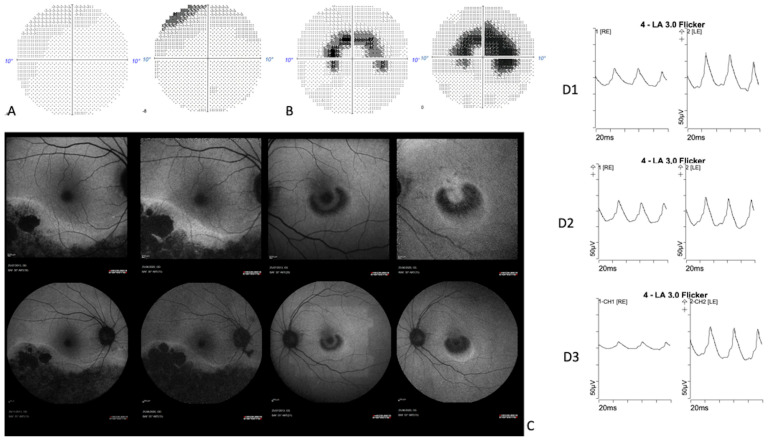
A seven-year follow-up comparison between: LE10-2 SITA fast Humphrey visual field (**A**); RE 10-2 SITA fast Humphrey visual field (**B**); 55° fundus autofluorescence, 32°fundus autofluorescence (**C**); flicker responses extracted from ISCEV full-field ERG in 2013 (**D1**), 2015 (**D2**) and 2020 (**D3**). The seven-year follow-up showed a slow progression of anatomical and functional alterations.

## Data Availability

Data sharing is not applicable to this article as no datasets were generated or analyzed during the current study. All patient data that we have access to are presented in the present study.

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
