# Peer review of "Extreme Interocular Asymmetry in an Atypical Case of a Hydroxychloroquine-Related Retinopathy"

_medicina, 2022, doi:10.3390/medicina58070967_

Round 1
Reviewer 1 Report
As you know, bull's eye maculopathy includes :
Age-related macular degeneration
Benign concentric annular dystrophy
Central areolar choroidal dystrophy
Chloroquine/hydroxychloroquine retinal toxicity
Chronic macular hole
Cone and cone-rod dystrophies
Stargardt disease
So you must check these differential diagnosis.
Furthermore clinical feature of a disease in the rt eye of your patient is different with typical HQ retinopathy. In order to suggest " this is the first reported case where such an asymmetry between the right and the left eye in the toxicity profile is observed" in your article, you need to confirm the underlying retinal disease like age-related macular degeneration.
Author Response
Response to reviewer 1
Comment 1: "As you know, bull's eye maculopathy includes ..."
We thank the reviewer for this suggestion. However we want to stress out that bull’s eye maculopathy is a bilateral disease characterized by a round, annular or ring-shaped lesion which relatively symmetric lesion between both eyes. In the present case, the macular lesion resembles bull’s eye lesion but was predominant inferiorly which is the classical picture of hydroxychloroquine and chloroquine toxicity. There was no family history of inherited retinal diseases and baseline retinal screening prior to treatment was normal excluding a preexisting retinal dystrophy. Fundus autofluorescence did not reveal additional alteration besides the loss of autofluorescence associated with atrophy, namely there was no flecks (Stargardt disease) and no drusen (age related macular degeneration (AMD). In addition, the age of the patient (50) would be highly unusual for AMD. Furthermore, the course of treatment and the hydroxychloroquine intake is compatible with drug toxicity. We have added a sentence in the text ‘Other retinal diseases associated with bull’s eye lesion were excluded due to normal pretreatment ophthalmic examination and multimodal retinal imaging (absence of flecks or drusen) and a predominant inferior atrophy typical of hydroxychloroquine toxicity.
Lally DR, Heier JS, Baumal C, Witkin AJ, Maler S, Shah CP, Reichel E, Waheed NK, Bussel I, Rogers A, Duker JS. Expanded spectral domain-OCT findings in the early detection of hydroxychloroquine retinopathy changes following drug cessation. Int J Retina Vitreous. 2016 Jul 18; 2:18. DOI: 10.1186/s40942-016-0042-y
Comment 2: "Furthermore clinical feature of a disease..."
We thank the reviewer for this remark. However, we refer the reviewer to previously published papers documenting HQ toxicity which we have quoted in our manuscript including recent descriptions of HQ toxicity in Asian subject more prone to peripheral involvement. As discussed above, normal pretreatment ophthalmic examination, the absence of a family history of inherited retinal disease, multimodal retinal imaging and the age of the patient are not suggestive of other retinal disease including AMD (50 year old subject and no drusen visible on OCT)
Reviewer 2 Report
It is an interesting subject on the hydroxychloroquine macular toxicity.
- The risk factors for hydroxychloroquine macular toxicity need to be further explained in discussion part.
- In addition, the cases that are at higher risk for this conflict should be identified and clearly mentioned how to screen them.
- In Figure 3, the descriptions should be marked in the images.
- Please more discussed about all phenotypes of HCQ retinopathy.
- The challenges of understanding of this pathophysiology of HCQ retinal toxicity should be more explained in the discussion part.
Author Response
- It is an interesting subject on hydroxychloroquine macular toxicity.
Thank you we believe it is worth sharing this case.
- The risk factors for hydroxychloroquine macular toxicity need to be further explained in the discussion.
Thank you for this comment. We added a paragraph listing AAO 2016 major risk factors for HCQ.
"The 2016 AAO recommendations defined by relying on a large cohort study of 2361 patients major risk factors for HCQ maculopathy.[1] These risk factors include an HCQ dose >5.0 mg/kg, treatment duration superior to 5 years, renal dysfunction, tamoxifen intake, and preexisting macular disease. [7]"
- In addition, the cases that are at higher risk for this conflict should be identified and clearly mentioned how to screen them.
We suggest that patients of Asian origin with significant risks of HCQ toxicity undergo a bi-annual screening of HCQ toxicity, including a Humphrey 10-2 and large SD-OCT vertical scans. Examiners must look critically at the edge of 10-2 fields. When possible, widefield FAF should be obtained, and 30-2 fields may be indicated in addition to 10-2 fields as recommended by Melles & all.
- In Figure 3, the descriptions should be marked in the images.
We updated Figure 3 with marks in the images. Thank you for this suggestion that increases the understanding of the image description.
- Please more discussed about all phenotypes of HCQ retinopathy.
We added this paragraph thanks to your comment :
To date, there are three main phenotypes of HCQ toxicity reported in the literature: parafoveal HCQ toxicity, which resembles a pseudo-bull's eye maculopathy predominant inferiorly; pericentral retinopathy that is mainly reported in patients of Asian origin but can also happen in patients of Caucasian origin and a severe phenotype that associates a parafoveal and pericentral phenotypes.
- The challenges of understanding this pathophysiology of HCQ retinal toxicity should be more explained in the discussion part.
Thank to your comment, we improved the discussion by adding this paragraph:
Otherwise, a recent article reports an asymmetric case of HCQ toxicity in a 62-year-old white woman with erosive rheumatoid arthritis and tried to address the challenges of understanding HCQ asymmetric retinal toxicity. She developed an
hydroxychloroquine toxicity in her phakic eye, with her aphakic fellow eye only mildly
affected. [21] Authors attributed this asymmetry to optical factors. In our case, the optical factor cannot explain the asymmetry. As discussed above, the classic genetic hypothesis or

Round 2
Reviewer 1 Report
Thank you for your revision.